# Overlapping *Streptococcus pyogenes* and *Streptococcus dysgalactiae* subspecies *equisimilis* household transmission and mobile genetic element exchange

Ouli Xie [1,2], Cameron Zachreson [3], Gerry Tonkin-Hill[4], David J. Price [1,5], Jake A. Lacey [1,6], Jacqueline M. Morris [6], Malcolm I. McDonald[7], Asha C. Bowen [8], Philip M. Giffard[9,10], Bart J. Currie [9,11], Jonathan R. Carapetis[8], Deborah C. Holt [9], Stephen D. Bentley [12], Mark R. Davies [6,14] & Steven Y. C. Tong [1,13,14] ✉

*Streptococcus dysgalactiae* subspecies *equisimilis* (SDSE) and *Streptococcus pyogenes* share skin and throat niches with extensive genomic homology and horizontal gene transfer (HGT) possibly underlying shared disease phenotypes. It is unknown if cross-species transmission interaction occurs. Here, we conduct a genomic analysis of a longitudinal household survey in remote Australian First Nations communities for patterns of cross-species transmission interaction and HGT. Collected from 4547 person-consultations, we analyse 294 SDSE and 315 *S. pyogenes* genomes. We find SDSE and *S. pyogenes* transmission intersects extensively among households and show that patterns of co-occurrence and transmission links are consistent with independent transmission without inter-species interference. We identify at least one of three near-identical cross-species mobile genetic elements (MGEs) carrying antimicrobial resistance or streptodornase virulence genes in 55 (19%) SDSE and 23 (7%) *S. pyogenes* isolates. These findings demonstrate co-circulation of both pathogens and HGT in communities with a high burden of streptococcal disease, supporting a need to integrate SDSE and *S. pyogenes* surveillance and control efforts.

*Streptococcus dysgalactiae* subspecies *equisimilis* (SDSE, commonly group C/G *Streptococcus*) is closely related to the better-known human pathogen, *Streptococcus pyogenes* (group A *Streptococcus*). SDSE shares much the same ecological niche on the human skin and throat as *S. pyogenes* and the two pathogens exhibit overlapping disease manifestations such as pharyngitis and invasive disease including necrotising fasciitis and streptococcal toxic shock syndrome[1].

Aboriginal communities in Australia are disproportionately affected by *S. pyogenes* disease including a high burden of impetigo, invasive disease, and post-infectious sequelae[2–4]. Drivers of disease critically include social determinants such as poor environmental conditions, household overcrowding, and access to healthcare[5]. The burden of disease caused by SDSE in these communities is not clearly defined. However, invasive SDSE disease also appears to disproportionately affect Aboriginal Australians[6]. In regions with a high burden of beta-haemolytic streptococcal disease and post-infectious sequelae, there has been evidence that superficial SDSE infection may trigger immune responses which cross-react with cardiac myosin[7,8].

---

These findings raise the possibility that SDSE may contribute to immune priming and the burden of rheumatic heart disease in those regions[7]. In high-income regions, emerging evidence has also described crude rates of invasive SDSE disease comparable to, and in some jurisdictions, greater than *S. pyogenes*, particularly in the elderly[6,9,10].

Whole genome comparisons of SDSE and *S. pyogenes* demonstrate extensive genomic homology including shared virulence factors such as the multi-functional surface M protein and evidence of horizontal gene transfer (HGT), frequently involving mobile genetic elements (MGEs)[11–13]. These similarities may contribute to shared disease phenotypes. Many *S. pyogenes* vaccine candidates are present in both species with evidence of cross-species homologous recombination[13].

Despite extensive genomic homology, there is in vitro evidence of possible cross-species competition. Strains of the two pathogens possess shared quorum sensing genes such as the *sil* locus with evidence of cross-species signalling[14]. Furthermore, anti-microbial peptides or bacteriocins such as SpbM/SpbN and the SDSE-specific Dysgalacticin, are found in some strains of SDSE and *S. pyogenes* with cross-species activity[15,16].

SDSE and *S. pyogenes* transmit by common pathways including respiratory droplets[1]. Recently, we have shown that asymptomatic *S. pyogenes* throat carriage is an important reservoir of transmission in high-endemic settings[17]. Transmission pathways of SDSE have not previously been described. Further, it is uncertain if in real-world studies transmission of one species competes with the other. In communities endemic for *S. pyogenes* infection with high rates of skin infection, rheumatic heart disease, and invasive disease, the current focus is largely on *S. pyogenes* control through skin sore and scabies control programs, and vaccine development. Understanding the transmission interactions of SDSE and *S. pyogenes* and anticipating the potential impact of disease control measures on cross-species behaviour is important to inform the design of surveillance programs and infection control efforts.

In this study, we examine the transmission of SDSE at a whole genome sequence (WGS) resolution using isolates collected in a household-based surveillance study over two years in two remote communities in the Northern Territory of Australia[18]. These transmission networks were compared to that of co-collected *S. pyogenes* isolates to assess for inter-species transmission interactions, and in the setting of co-circulation, their genomes were systemically examined for evidence of cross-species HGT of MGEs carrying key virulence and antimicrobial resistance genes.

## Results

### Sampling and clinical epidemiology

Two remote Aboriginal communities in the Northern Territory of Australia were prospectively followed for a two-year period between 2003 and 2005[18,19]. Observations for one community (community 3) commenced in June 2004 as it replaced an initial community (community 2) with low recruitment. Communities 1 and 3 were included in this study.

Households (18 in community 1 and 20 in community 3) were visited approximately monthly, allowing for access affected by weather and cultural events (Supplementary Fig. 1). At each visit, throat swabs were taken regardless of symptoms and skin swabs were taken from impetigo lesions. From a total of 4547 person consultations during 486 household-visits, 1087 individuals (547 from community 1 and 540 from community 3) were sampled from which 330 SDSE isolates (252 from community 1 and 78 from community 3) were recovered. Of the 330 isolates, 8 were from skin, and 322 from throat swabs of which only one case reported a sore throat. *S. pyogenes* was recovered on 327 occasions (218 from community 1 and 109 from community 3) with 208 isolated from throat swabs and 119 from impetigo lesions. The age distribution of individuals carrying SDSE and *S. pyogenes* was similar[18,19]. SDSE was isolated on 33

occasions from children <5 years of age, on 82 occasions from children between 5 and 10 years, on 112 occasions from children between 10 and 15 years, and on 103 occasions from individuals aged ≥15 years compared to 46, 98, 104, and 79 cases for *S. pyogenes*, respectively. Detailed descriptions of the epidemiology of cases were described previously[18,19].

There was a high rate of individual mobility in and out of households with a median of 28 people (range 6–57) enrolled per household over the study period. Each individual was observed at a median of 3 visits (range 1–19, intermittently sampled); and as such, duration of carriage in individuals could not be determined. Households were positive (i.e. at least one individual positive) for SDSE for a median of 56 days (95% CI 31–65 days) and then re-acquired SDSE a median of 37 days later (95% CI 34–54 days).

### Whole genome sequencing reveals detailed transmission clusters

From the 330 SDSE isolates, 294 (89%) were recovered for WGS and passed quality control (Supplementary Data 1a). All sequenced isolates were predicted to carry at least one bacteriocin biosynthetic cluster including 267 (91%) with *spbMN* encoding SpbM/SpbN which has previously been shown to be inhibitory to SDSE and *S. pyogenes*[15] (Supplementary Data 1b). The *sil* locus was present in 235 (80%) of isolates. All 12 *S. pyogenes* vaccine candidate genes (*adi, fbp54, oppA, pulA, scpA, shr, slo, spy0762, spy0942, spyAD, srtA, ropA*) which were previously found in >99% of SDSE and *S. pyogenes* global isolates[13] were also present in all sequenced SDSE isolates from this study. Sequencing and analysis of 315/327 (96%) *S. pyogenes* isolates recovered from communities 1 and 3 were reported previously[17]. All *S. pyogenes* isolates were predicted to carry at least one bacteriocin biosynthetic cluster including 290 (92%) with *spbMN* (Supplementary Data 1c). The *sil* locus was present in 128 (41%) of *S. pyogenes* isolates.

Using traditional epidemiological markers, *emm* type and multi-locus sequence type (MLST), the SDSE isolates represented 19 *emm* types (23 *emm* subtypes), 21 MLSTs, and 26 *emm*-MLST combinations (Supplementary Data 1a). Of these, 8/26 (31%) *emm*-MLST groups were found across both communities.

To determine a WGS threshold for clustering of SDSE strains, we examined genomic variation of isolates of the same strain found longitudinally on multiple occasions from the same individual. Intra-host variation was used to predict longitudinal diversity of strains forming transmission chain as well as technical variations in single nucleotide polymorphism (SNP) calling. SDSE was found in 58 individuals on more than one occasion including three who were positive on five occasions, four on four occasions, 15 on three occasions, and 36 on two occasions. Using *emm* and MLST as markers, 36 individuals had the same strain on more than one occasion including six individuals with the same isolate on three occasions, one individual on four occasions, and one on five occasions (Supplementary Fig. 2). Pairwise SNP distances were calculated between these isolates and a threshold of <8 SNPs was determined for WGS transmission clustering (Supplementary Fig. 3).

Phylogenetic reconstruction supported 18 distinct SDSE lineages/global genomic sequence clusters[13] present across both communities (Fig. 1). High-resolution genomic transmission clusters based on single linkage clustering at a SNP threshold of <8 and >99% shared gene content, revealed much finer detail than the traditional epidemiological markers (Fig. 2). A total of 37 SDSE transmission clusters representing 237 (81%) isolates were inferred with an additional 57 singleton isolates (Supplementary Data 1a). Transmission clusters were supported by core SNP phylogenies and presence-absence of virulence and/or antimicrobial resistance genes (Supplementary Fig. 4a–c) with significant diversity within *emm* types (Fig. 2) and evidence of MGE gain/loss events carrying antimicrobial resistance and/or virulence factor genes among closely related isolates.

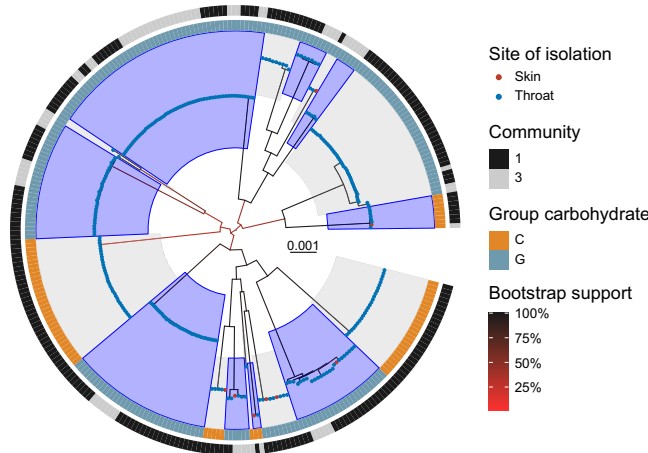

**Fig. 1 | Maximum-likelihood phylogeny of 294 *Streptococcus dysgalactiae* subsp. *equisimilis* (SDSE) isolates from 47,432 parsimony-informative sites.** Sequences were aligned against SDSE reference genome GGS_124 (NC_012891.1) with mobile genetic element regions masked. Distinct genomic sequence clusters determined by PopPUNK[31] as previously defined by a global SDSE dataset[13], are denoted by alternating blue and grey highlights from internal nodes. Site of isolation is coloured by blue (throat) and red (skin) tips. The inner ring denotes the Lancefield group carbohydrate and the outer ring the community of isolation. Bootstrap supports are shown as branch colour gradients and were calculated using the ultrafast bootstrap approximation demonstrating some uncertainty in deep branches of the phylogeny[37]. Scale bar represents substitutions per site.

The two largest transmission clusters consisted of 32 isolates each and clusters with four or more isolates made up a total of 204 isolates (69%). Transmission clusters were present across a mean of 3 households (range 1–16). Despite the finding of eight *emm*-MLST groups across both communities, the WGS analysis indicated that only a single transmission cluster spanned both communities. The upper limit of the pairwise SNP distance between isolates of the same transmission cluster was 16 SNPs (median 4) compared to 791 SNPs (median 20) within the same MLST, 5491 SNPs (median 25) within the same *emm* type, 638 SNPs (median 19) within the same *emm*-MLST combination, and 1505 SNPs (median 21) within the same genomic sequence cluster (Supplementary Fig. 5), highlighting the limitations of other markers in determining recent transmission clusters.

There was no clear pattern of *emm* type replacement of SDSE isolates over time in the two communities in contrast to sequential replacement of *S. pyogenes emm* types as reported previously[17,18]. Consistent with this finding, SDSE transmission clusters persisted for longer in the two communities (median of 349 days, 95% CI 189–440 days) compared to *S. pyogenes* (median of 241 days, 95% CI 181–259 days, log-rank *p* = 0.009) (Supplementary Fig. 6).

**Network analysis supports independent household transmission dynamics for SDSE and *S. pyogenes***

SDSE transmission between households within each community was modelled by inferring links between isolates of the same transmission cluster detected at successive community visits (transmission window 12–44 days), including intra-household transmission events. Individuals were grouped by household which formed the nodes of the transmission work. Analysis of the transmission network revealed 123 SDSE putative transmission edges (events) in community 1 and 14 edges in community 3, which had a shorter duration of sampling and fewer isolates detected (Table 1). Consistent with the low number of SDSE isolates from skin sores, all but one transmission edge was attributed to isolates from throat swabs for SDSE in contrast to 50/173 (29%) edges attributed to a predicted skin source for *S. pyogenes*.

To test the hypothesis that transmission of SDSE or *S. pyogenes* may interfere with transmission of the other species, the overlap between inferred transmission networks of the two species was compared to a null model in which any cross-species interaction was removed. Transmission overlap was defined as the proportion of inferred SDSE transmission edges that corresponded to an inferred transmission of *S. pyogenes*. An overlapping edge corresponded to transmission of both SDSE and *S. pyogenes* which occurred between the same pair of households within the same transmission window without distinguishing which household acted as source. To generate a null model of transmission overlap, household labels in the inferred SDSE transmission network were randomised while preserving the *S. pyogenes* network. This process preserves important structural features of the SDSE network including degree distribution, and any clustering of SDSE transmission between households, while removing any direct cross-species effects related to the transmission of *S. pyogenes*.

Overlaying the transmission networks of the two species found a highly interconnected network with 11 shared transmission edges—nine in community 1 and two in community 3 (Fig. 3). The number of shared edges in each community was consistent with the distribution under the null model providing no evidence of inter-species transmission interference (one-sided *p*-value ≤ observed value for community 1 = 0.75, community 3 = 0.94) (Supplementary Fig. 7a, c). Results were similar when restricting the analysis to isolates only from throat swabs (Supplementary Fig. 7b, d). These results indicate no evidence of an interaction between the two species in their household transmission patterns.

Although only 11/137 (8%) of total SDSE transmission edges were shared with *S. pyogenes*, the combined transmission networks demonstrated extensive crossover of the two organisms at the household level—SDSE and *S. pyogenes* co-occurred in the same household on 100/486 (21%) of household-visits (Fig. 4). To infer a null model of co-occurrence of SDSE and *S. pyogenes* in households while removing cross-species transmission effects, SDSE and *S. pyogenes* positive swabs were randomised across all swabs at each community visit. To account for grouping of isolates within households, isolates from the same transmission cluster were collapsed to a single positive result during the same household visit. The observed co-occurrence of SDSE and *S. pyogenes* within households was consistent with the model of independent inter-species transmission without evidence of interference (one-sided *p*-value ≤ observed value across both communities = 0.62). Results from a sensitivity analysis limited to isolates from throat swabs were consistent (Supplementary Fig. 8a, b).

In a subset of swabs, more than one large colony beta-haemolytic *Streptococcus* colony variant was present on culture and examined. On 15 occasions in 14 individuals (13 throat, 2 skin), SDSE and *S. pyogenes* were isolated on the same swab. There was no *emm* type or lineage restriction for co-occurring isolates and both *sil* positive and *sil* negative strains were detected. Accounting for simultaneously collected skin and throat swabs, co-occurrence in the same individual occurred on 26 occasions in 24 individuals. These findings suggest that consistent with independent transmission at the household level, SDSE and *S. pyogenes* can also co-occur within the same individual, including at the same site. As multiple colonies were tested only if there were colony variants and the two organisms are generally considered indistinguishable morphologically, this only represents a lower bound of co-occurrence within individuals.

**Co-occurrence of SDSE and *S. pyogenes* facilitates shared mobile genetic elements**

We have previously demonstrated extensive genomic overlap between SDSE and *S. pyogenes* in the context of global genome databases[13]. In the setting of extensive household co-occurrence of the two species, we sought to find evidence of shared MGEs between the two species. Using a pangenome synteny-based approach, MGEs were systemically extracted from both SDSE and *S. pyogenes* isolates and examined for elements with >99% nucleotide identity across species[13,20].

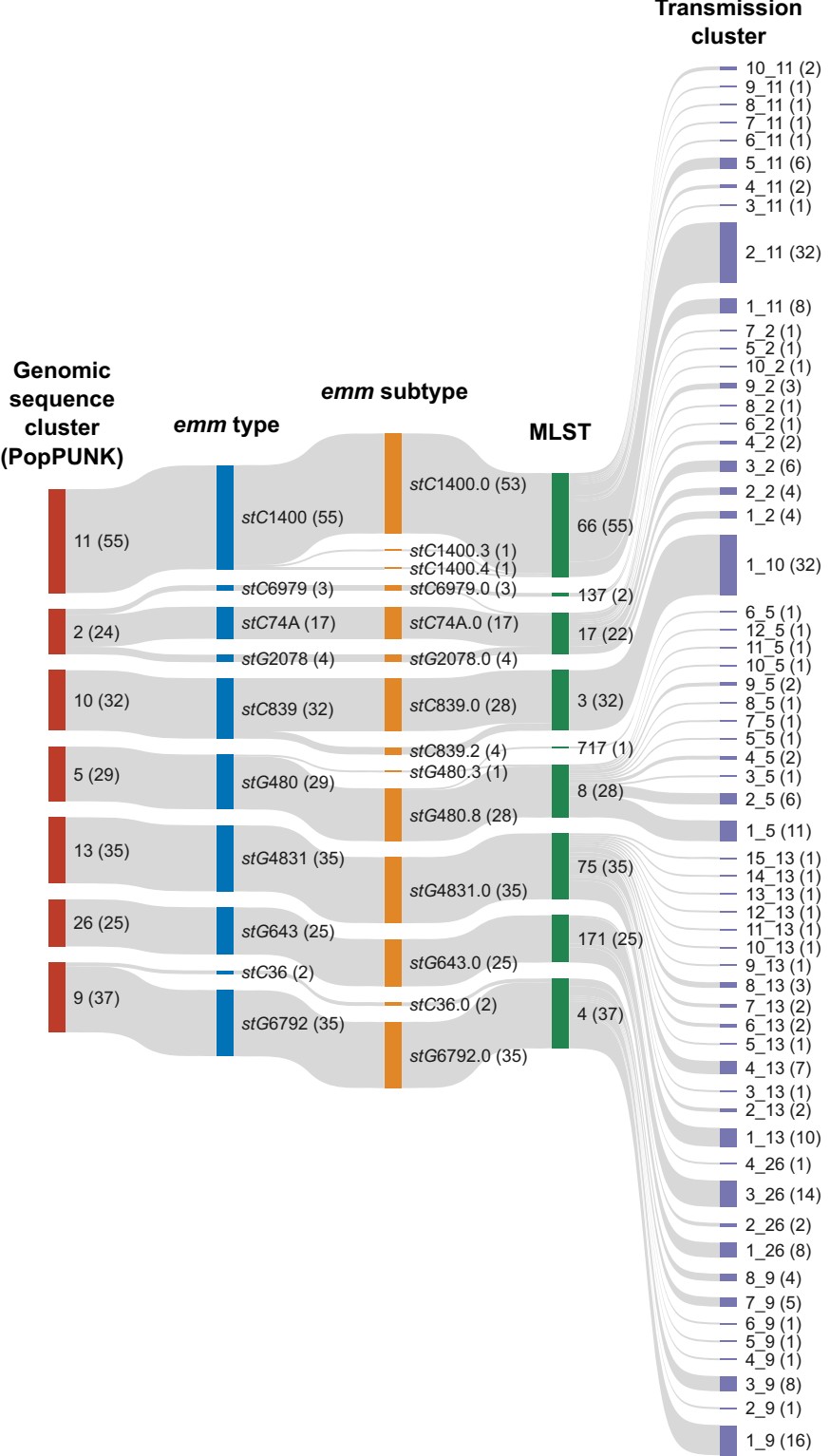

**Fig. 2 | Alluvial plot of the relationship between the largest 7/18 *Streptococcus dysgalactiae* subspecies *equisimilis* (SDSE) genomic sequence clusters (representing 237/294 isolates) as determined by PopPUNK against *emm* type, *emm* subtype, multilocus sequence type (MLST) and transmission clusters determined using single linkage clustering at a SNP threshold of 99% shared gene content.** The number of isolates in each category is denoted within brackets. From 10 *emm* types (14 *emm* subtypes) and 9 MLSTs shown, 60 transmission clusters were determined. Traditional markers such as *emm* subtype in some cases oversplit SDSE clusters as demonstrated by *stC*839.0 and *stC*839.2 which differ by only one SNP within their hypervariable *emm* region and otherwise fall within the same transmission cluster.

Three near-identical MGEs were found to be present in SDSE and *S. pyogenes* with variable presence across closely related isolates with as few as 0–11 core SNPs, suggestive of recent MGE gain/loss events within each of these strains (Fig. 5a). A 53 kbp prophage, φ1207.3[21], carrying *mef*(A)/*msr*(D) macrolide efflux resistance genes was carried at a conserved cross-species genomic location (between SDEG_RS07105 and SDEG_RS07110 in reference genome GGS_124 NC_012891.1) and was present in 5/8 *emm*58.8 ST549 *S. pyogenes* and 31 SDSE isolates across subsets of five different lineages (Fig. 5b). The φ1207.3 prophage was variably present in closely related isolates in both SDSE and *S. pyogenes* with ancestral state reconstruction

indicating likely recent acquisition within at least two lineages (*stG*4831 SDSE and *emm*58 *S. pyogenes* lineages) rather than being carried lineage-wide (Supplementary Fig. 9a–c).

A second prophage, φMGAS5005.3 carrying the streptodornase gene *sda1*, previously described to be shared across species, was also found in a cross-species conserved insertion region[13]. φMGAS5005.3 was present in 14/16 *S. pyogenes emm*1.0, ST28 isolates with a maximum of 3 SNPs between isolates and 16 SDSE isolates across subsets of two different lineages. The φMGAS5005.3 is known to have been acquired ancestrally to modern *emm*1 *S. pyogenes* strains[22]. In contrast, acquisition towards the tips of the phylogeny was supported by ancestral state reconstruction in SDSE *stC*1400 isolates (Supplementary Fig. 9d). Inference in *stG*6792 isolates was ambiguous due to near-perfect separation along two basal branches of the lineage phylogeny.

An 18 kbp integrative conjugative element (ICE)-like segment carrying the tetracycline resistance gene, *tet*(M), was present in four *S. pyogenes* and eight SDSE isolates at three distinct insertion regions (Fig. 5c, Supplementary Fig. 10). Ancestral state reconstruction was not informative as the ICE-like element was carried lineage-wide when present.

At least one of these MGEs which carried antimicrobial resistance-associated genes or virulence-encoding genes, was found in 55 (19%) of SDSE and 23 (7%) of *S. pyogenes* isolates. SDSE isolates carrying these shared MGEs were found across both communities while *S. pyogenes* isolates carrying shared MGEs were restricted to single communities (community 1 for φ1207.3, community 3 for φMGAS5005.3 and the

**Table 1 | Number of inferred unweighted transmission edges between households at successive community visits in two remote communities**

| | Household | | Community | | |
|---|---|---|---|---|---|
| | All isolates | Throat source | All isolates | Throat source | Total |
| *Streptococcus dysgalactiae* subspecies *equisimilis* | | | | | |
| Community 1 | 22 (18%) | 22 (18%) | 101 (82%) | 100 (81%) | 123 |
| Community 3 | 3 (21%) | 3 (21%) | 11 (79%) | 11 (79%) | 14 |
| *Streptococcus pyogenes* | | | | | |
| Community 1 | 15 (11%) | 9 (7%) | 116 (89%) | 75 (57%) | 131 |
| Community 3 | 4 (10%) | 3 (7%) | 38 (90%) | 36 (86%) | 42 |

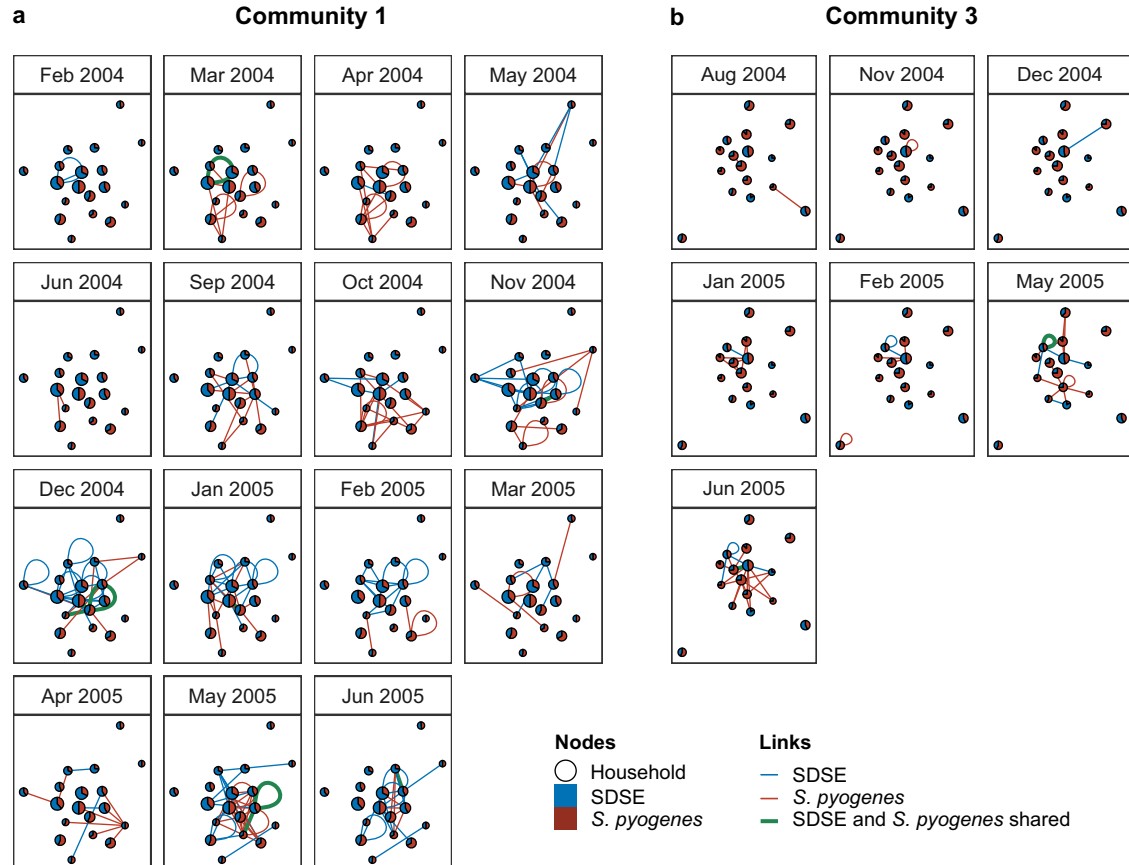

**Fig. 3 | Transmission links between households across consecutive community visits.** Transmission links for Community 1 (**a**) and Community 3 (**b**). Households are represented by nodes proportional in size to the number of participants enrolled at each household and coloured by the proportion of *Streptococcus pyogenes* (red) and *Streptococcus dysgalactiae* subsp. *equisimilis* (SDSE, blue) isolates detected in the household across the entire study period. Transmission links are represented by undirected and unweighted edges between households and coloured by species with shared edges highlighted in green. Loops correspond to predicted transmission edges between unique individuals within the same household. Only community visits where transmission edges were predicted are shown. Source data are provided as a Source Data file.

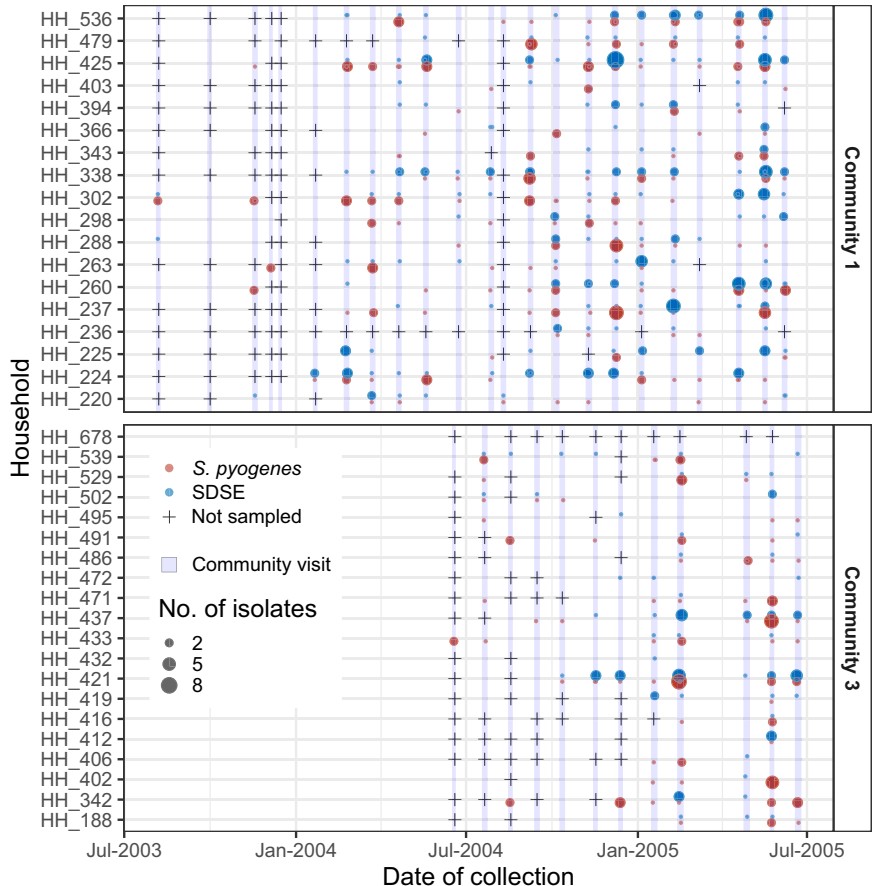

**Fig. 4 | Co-occurrence of *Streptococcus dysgalactiae* subsp *equisimilis* (SDSE) and *Streptococcus pyogenes* in households in communities 1 and 3 at each community visit (light blue highlights) during the study period.** Detection of SDSE (blue) and *S. pyogenes* (red) are denoted by points with the size of each point proportional to the number of isolates. Community visits where a household was not sampled are denoted by crosses. SDSE and *S. pyogenes* co-occurred on 100/486 (21%) of household-visits. Source data are provided as a Source Data file.

ICE-like element). There was no clear pattern of *emm* type or lineage restriction for shared MGEs.

While directionality of MGE movement could not be inferred, including distinguishing between inter-species versus intra-species dissemination or exclusion of acquisition from a third species, the presence of near-identical elements at conserved insertion regions, suggests that overlapping transmission may facilitate shared MGEs from a common pool. The carriage of these MGEs across multiple distinct lineages suggests that these shared MGEs may lead to dissemination of antimicrobial resistance and virulence-associated genes.

## Discussion

Using WGS-level resolution, we were able to reconstruct SDSE household transmission networks and compare it to co-collected *S. pyogenes* isolates, demonstrating extensive co-circulation in remote Aboriginal communities with a high burden of streptococcal disease. Despite occupying similar niches on the skin and throat, we show that the two organisms transmit independently without evidence of interference at the household level. In the setting of extensive transmission cross-over in households, we find multiple MGEs present across both populations carrying antimicrobial resistance or virulence factor genes with evidence suggestive of recent gain/loss events. This analysis of a dataset of densely co-sampled SDSE and *S. pyogenes* isolates provides a level of transmission detail and examination of real-world inter-species transmission dynamics and HGT which to our knowledge, has not previously been described for beta-haemolytic streptococci.

SDSE is increasingly being recognised as an important cause of invasive human disease with recent studies suggesting incidence and mortality comparable to *S. pyogenes*[6,9,10]. While not traditionally considered as a cause of acute rheumatic fever/rheumatic heart disease (ARF/RHD), reports from northern Australia suggest that at least in high-incidence areas of ARF/RHD, SDSE throat carriage may have the potential to induce cardiac myosin cross-reactive antibodies mimicking that seen with *S. pyogenes*[7,8]. Although our findings represent transmission dynamics in a high disease-burden setting where social determinants such as household overcrowding and access to healthcare may be strong drivers of transmission, they are also relevant for other disadvantaged settings globally with the highest burden of streptococcal disease. Therefore, the finding of extensive throat transmission of SDSE, including persistence of transmission clusters longer than that of *S. pyogenes*, underscores a need to understand further its contribution to disease including its role in immune priming for ARF/RHD which in turn has important disease control implications.

Additionally, interactions between SDSE and *S. pyogenes* such as HGT and homologous recombination are key drivers in bacterial population dynamics, and may influence *S. pyogenes* and SDSE biology[13]. Notably, genes encoding antigens currently under investigation as *S. pyogenes* vaccine candidates are frequently also found in SDSE[13]. Our findings of extensive household co-occurrence may provide an opportunity for HGT which we demonstrate in the setting of shared MGEs. We show three near-identical MGEs were present across different lineages in SDSE and *S. pyogenes* including presence and absence in closely related isolates suggestive of recent gain/loss

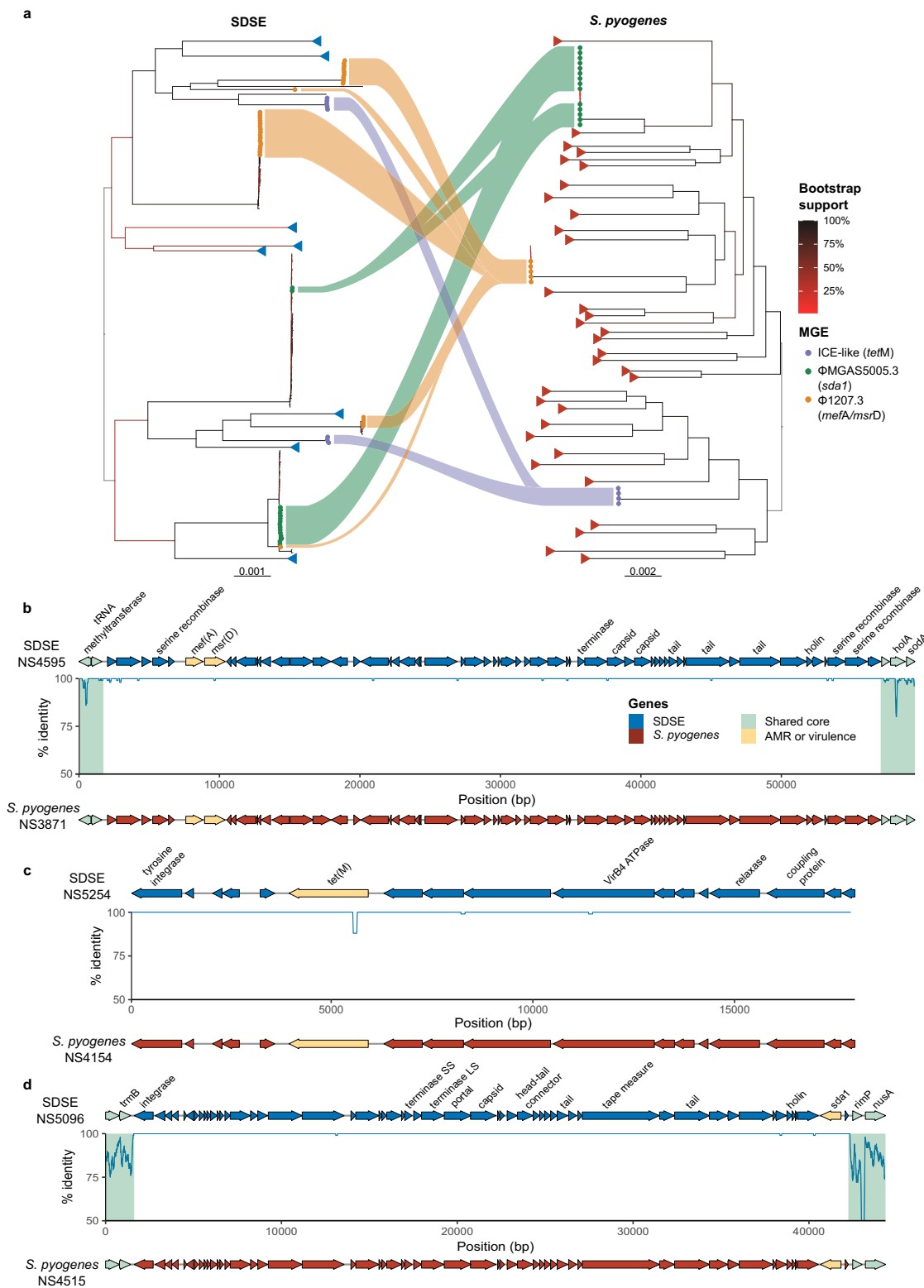

events. Recent acquisition of MGEs in some lineages was supported by ancestral state reconstruction. These MGEs carried antimicrobial resistance and virulence genes such as the macrolide efflux genes *mef*(A)/*msr*(D), tetracycline resistance *tet*(M), and the streptodornase gene *sda1*. While we cannot infer directionality of HGT of MGEs across species compared to intra-species dissemination or acquisition from an intermediary species (e.g. the ICE-like Tn916 MGE has also been

detected at >99% nucleotide identity in other organisms such as *Staphylococcus aureus*), at least one of these MGEs was present in 55 (19%) of SDSE and 23 (7%) of *S. pyogenes* isolates. This underscores the importance of integrating SDSE with *S. pyogenes* surveillance as we seek to improve our understanding of transmission and disease pathogenesis of the two organisms and as efforts move towards a possible *S. pyogenes* vaccine which may introduce selection pressures

**Fig. 5 | Shared mobile genetic elements (MGE) across *Streptococcus dysgalactiae* subsp. *equisimilis* (SDSE) and *Streptococcus pyogenes* isolates.**
**a** Maximum likelihood trees of SDSE and *S. pyogenes* with isolates carrying three near-identical (>99% nucleotide identity) MGEs highlighted by tree tip points. Genomic sequence clusters without the three MGEs of interest are collapsed and denoted by blue (SDSE) and red (*S. pyogenes*) triangles at tree tips. Flows link corresponding shared MGEs across the species but do not imply directionality of transfer. The *emm* sequence type of genome sequence clusters sharing MGEs are labelled for each species, respectively. Bootstrap supports are shown as branch colour gradients and were calculated using ultrafast bootstrap approximation[37]. Scale bars represent substitutions per site. **b** A 54 kbp prophage, φ1207.3, carrying macrolide efflux resistance genes *mef*(A) and *msr*(D) was present with >99.9% nucleotide identity across SDSE and *S. pyogenes*. A representative SDSE element

from isolate NS4595 was aligned against a representative *S. pyogenes* sequence (NS3871) with percentage nucleotide identity calculated using Hamming distance and plotted in 100 bp sliding windows. The element was present in a cross-species conserved insertion region with flanking core genes highlighted in green. **c** An 18 kbp integrative conjugative element (ICE)-like MGE carrying the tetracycline resistance gene, *tet*(M), was present with >99.9% nucleotide identity across species. The element was present at three different genomic insertion regions and thus flanking core genes are not shown. In the example shown, a 12 bp in-frame deletion was present at the 5' end of *tet*(M) in the SDSE element which was distant from the active ribosomal binding domain. **d** A 41 kbp prophage φMGAS5005.3 carrying the streptodornase gene *sda1*, was shared across species with >99.9% nucleotide identity at a cross-species conserved insertion region as has been described previously[13]. Source data are provided as a Source Data file.

across both organisms. Further integration of cross-species surveillance may also shed light on biological factors which may facilitate or restrict cross-species HGT including differences in CRISPR arrays, restriction–modification systems, and phage receptors.

SDSE and *S. pyogenes* occupy similar ecological niches in the throat and on the skin with overlapping disease manifestations such as pharyngitis. Cross-species interaction and competition have been demonstrated such as the expression of bacteriocins which are able to inhibit the other species and cross-species quorum sensing involving the two-component regulator, SilAB with its signalling peptide SilCR[14–16]. However, the *sil* locus and characterised bacteriocins such as Dysgalacticin and SpbM/SpbN are variably present in SDSE and *S. pyogenes* and it is unclear if in vitro interactions translate to real-world transmission dynamics. Our data demonstrate that despite evidence of possible in vitro interference, SDSE and *S. pyogenes* appear to transmit independently with highly interconnected household transmission networks in a high-burden setting.

SDSE was almost exclusively isolated from the throat in this study[18]. The mechanism behind the predilection for the throat for SDSE in comparison to the wider presence of *S. pyogenes* across throat and impetigo lesions is unclear. The age of individuals included in this study with SDSE was not different to those with *S. pyogenes* with the highest prevalence in 10–15-year-olds and does not explain the throat predominance of SDSE[18,19]. Despite the genomic similarities between SDSE and *S. pyogenes*, their virulence repertoires differ with many *S. pyogenes* virulence genes infrequently found in SDSE and may potentially underly phenotypic differences[13]. Cross-species genotype-phenotype associations could not be drawn from this study due to the near-perfect separation between skin and throat sites for SDSE. However, sensitivity analyses restricting cross-species transmission analyses to throat isolates were concordant with the primary analysis without any evidence of cross-species interference.

Despite evidence of independent transmission at a household level, with household co-occurrence of SDSE and *S. pyogenes* on 100/486 (21%) of household-visits, the frequency of presence of SDSE and *S. pyogenes* in the same swab is unclear. SDSE and *S. pyogenes* are both large colony, beta-haemolytic streptococci and are generally indistinguishable by colony morphology. Given only representative colonies were characterised in this study, the true frequency of co-colonisation of SDSE and *S. pyogenes* in the same individual could not be estimated other than on 15 occasions when colony variants were present. In fact, this is a common limitation of carriage studies to date seeking to determine the prevalence of SDSE and *S. pyogenes* from throat swabs[23–25]. Given our findings of household-level transmission dynamics, future studies should consider methods such as WGS from plate sweeps or deep sequencing of swabs to determine co-occurrence in individuals. These methods have also previously been shown to improve resolution of intra-host diversity and reconstructing transmission and may offer greater insight into cross-species transmission dynamics[26].

Our study has some limitations. This study was carried out in a remote and tropical setting in northern Australia in Aboriginal

communities with a high burden of *S. pyogenes* disease including impetigo, ARF/RHD and invasive disease. Therefore, transmission dynamics, co-occurrence of the two organisms, and even frequency of HGT may be influenced by sociodemographic factors and differ in temperate, high-income regions. There was a high level of population mobility in and out of households in these communities and thus individual level transmission dynamics and duration of carriage could not be determined due to limited longitudinal sampling of most individuals. Sampling more frequently than monthly may also provide greater resolution for transmission including attribution of transmission source. Additionally, while SDSE was only found from eight impetigo/skin sore swabs, intact skin was not sampled. Therefore, it is unclear if SDSE on healthy skin may contribute to transmission.

In summary, this study demonstrates important transmission dynamics of SDSE and *S. pyogenes*. The two closely related pathogens frequently co-occur within households with interconnected transmission networks but without evidence of inter-species interference across households in a high disease-burden setting. Transmission overlap and shared niches, particularly in the human throat, may facilitate interspecies gene flow including clinically important determinants such as antimicrobial resistance genes. These findings emphasise a need to further understand the interactions between these pathogens including in the context of ARF/RHD in high burden regions. The immunopathogenesis of ARF remains poorly understood despite many decades of research and the specific events antecedent to each episode of ARF are elusive with respect to the role of *S. pyogenes* in skin lesions and SDSE in the throat. As interventions targeting *S. pyogenes* take place, it is possible that SDSE may also be affected. That impact could potentially be a reduction in SDSE disease (e.g. by vaccines that may target common antigens) or conversely by SDSE filling an ecological niche if *S. pyogenes* infection or carriage is selectively targeted (e.g. in primary care interventions that expand the use of *S. pyogenes* rapid diagnostics for throat swabs). Incorporating research, surveillance, and control efforts of SDSE with *S. pyogenes* will improve the understanding of both pathogens individually and cross-species interactions in relation to clinical disease burden, disease phenotypes, and future response to vaccine interventions.

## Methods
### Isolate collection and culture
The current study received ethics approval from the Human Research Ethics Committee of the Northern Territory Department of Health and Menzies School of Health Research (approval 2015-2516). Informed consent was obtained as part of the original surveillance study and renewed informed written consent was waived for this study as no new data was collected from participants. The original study was approved by the local ethics committee together with the community, family, and individual consent for study procedures[18,19].

Isolates were collected from a previously reported prospective surveillance study in three remote Aboriginal communities in remote Northern Territory, Australia, which were visited approximately

monthly over a two-year period from August 2003 to June 2005[18]. At each visit, researchers collected throat swabs regardless of symptoms from participants and examined for skin sores both purulent and dry, which were also swabbed. Due to high population mobility, individuals were identified as part of households for analyses, including family groups residing in one or two adjacent houses.

Swabs were inoculated onto horse blood agar and selective media containing colistin and nalidixic acid and transported for culture at a central laboratory in Darwin, Australia. Plates were incubated at 37 °C in 5% $CO_2$ and examined after 24 and 48 h. A single representative large colony beta-haemolytic isolate was selected for typing (Streptococcal Grouping Kit, Oxoid Diagnostic Reagents) unless significant differences in colony morphology and/or haemolysis intensity was observed, in which case additional colonies were also selected.

### Whole genome sequencing and typing

Lancefield group C/G streptococcal isolates were retrieved from stored glycerol stocks kept at −70 °C. Microbial DNA was extracted and 150 bp paired-end libraries were sequenced using the HiSeq X Ten platform (Illumina, San Diego, CA, USA). Fifty-four SDSE sequences were previously published by ref. 13. *S. pyogenes* sequences were previously described by ref. 17 and available under BioProjects PRJNA879913 and PRJEB2232. One isolate was not able to be retrieved for sequencing.

Genomes were assembled using Shovill v1.1.0 (https://github.com/tseemann/shovill) with SPAdes assembler v3.14.0[27]. Only assemblies with <150 contigs (mean 102, range 66–131), total assembly size between 1.9 and 2.4 Mb (mean 2.09 Mb, range 1.91–2.29 Mb), and GC% between 38% and 40% (mean 39.4%, range 39.0–39.8%) were included. Annotations were generated using Prokka v1.14.6[28]. Reads from Lancefield group C/G streptococcal isolates were checked for contamination and confirmed to be SDSE using Kraken2 v2.1.2[29]. Any sequences with >5% reads assigned to a species other than SDSE, with the exception of *S. pyogenes* where up to 15% was allowed due to high genetic homology, was excluded. Additional measures to exclude contamination included no MLST matches to an *yqiL* allele which is present in *S. pyogenes* but not SDSE, detection of only a single *emm* allele, and concordance of *emm* type with the original surveillance study[18].

In silico typing of the hypervariable N-terminal domain of the *emm* gene was performed using emmtyper v0.2.0 (https://github.com/MDU-PHL/emmtyper) and MLST assigned using MLST v2.22.0 (https://github.com/tseemann/mlst)[30]. Genomic sequence clusters, representative of global SDSE populations, were assigned using PopPUNK v.2.60 with a scheme available at https://www.bacpop.org/poppunk/ (v1)[13,31]. Antimicrobial resistance, virulence and vaccine candidate genes, and the presence of the *sil* locus were inferred as previously described[13]. Bacteriocin biosynthetic gene clusters (RiPP-like, lanthipeptide class I, lanthipeptide class II) were inferred using antismash v7.1.0[32] in addition to the presence of *spbMN* at 70% amino acid identity and length. Genome metadata is available in Supplementary Data 1. *S. pyogenes* genomic sequences clusters were assigned with a scheme available at https://poppunk.net/pages/databases.html[33].

The pangenomes for SDSE and *S. pyogenes* were constructed using Panaroo v1.2.10[34] in 'strict' mode with initial clustering at 98% length and sequence identity followed by a family threshold of 70%. Core genes were defined as genes present in ≥99% of genomes. Pangenome gene synteny was mapped using Corekaburra v0.0.5[35].

Maximum likelihood phylogenetic trees for SDSE and *S. pyogenes* isolates were inferred using IQ-tree v2.0.6 with a GTR + F + G4 model and 1000 UFBoot replicates[36,37]. Alignments for SDSE were generated using Snippy v4.6.0 (https://github.com/tseemann/snippy) against reference genome GGS_124 (NC_012891.1) and *S. pyogenes* against reference genome MGAS5005 (NC_007297.2) with MGE regions masked. Recombination was not masked. Maximum parsimony trees

were inferred within genomic sequence clusters to validate predicted transmission clusters using phangorn v2.10.0 and SNP alignments generated by split kmer analysis (SKA v1.0)[38,39].

### MGE comparison

MGEs were systemically extracted from SDSE and *S. pyogenes* genomes using a pangenome synteny-based approach based on proMGE and used previously to examine global SDSE and *S. pyogenes* datasets[13,20]. Briefly, segments of accessory genes were extracted and searched for presence of recombinase/integrase genes and ICE/prophage structural genes. Based on the carriage of different recombinase/integrase families, accessory segments were classified into MGE classes. The pipeline was modified to extract nucleotide sequences of accessory genomic segments classified as MGEs. Sequences from SDSE and *S. pyogenes* were initially clustered using CD-HIT[40] v4.8.1 with a sequence identity threshold 0.8 and length difference cut-off 0.8. Clusters with sequences from both species were inspected and pairwise alignments generated using minimap2 v2.24[41]. To infer the presence/absence of MGEs shared across species in ancestral nodes of the phylogeny, ancestral character state estimation was performed using ape v5.7[42] with an equal rates model. As the aim was to infer gain/loss events close to the tips of the tree rather than lineage-wide characteristics, estimation was performed using maximum likelihood phylogenetic trees inferred from SKA alignments generated separately for each global genomic sequence cluster.

### Transmission clustering

Transmission clusters representing isolates predicted to have formed a recent transmission chain based on WGS data was determined using a SNP threshold of <8 and >99% shared gene content. A SNP threshold of <8 was determined using the maximal SNP distance between isolates of the same *emm* and MLST isolated within the same individual, a surrogate for SNP diversity within a single infecting strain. A gene content threshold of >99% determined from pangenome analysis was chosen to capture MGE gain/loss events.

Pairwise SNP distance between isolates within the same genomic sequence cluster was performed using SKA v1.0[38] from reads using the 'fastq' command with default coverage cut-off 4, minimum minor allele frequency 0.2, minimum base quality 20 and kmer size 15. Single linkage clustering with a SNP distance of <8 was calculated using the 'distance' command. SKA requires exact kmer matches in order to detect SNPs between the flanking/split kmers and therefore may miss SNPs between more divergent sequences. The 150 bp hypervariable N-terminal region of the *emm* gene poses such a challenge and has previously been shown to be able to undergo recombination. As such, isolates clustered by SKA were checked for matching *emm* types. In the case of different *emm* subtypes, alignments of the hypervariable region *emm* region were manually inspected to determine the number of SNPs. Finally, a pangenome comparison and single linkage clustering within each SKA cluster at 20 genes (approximates 99% gene similarity) was performed to generate the final transmission clusters.

Transmission cluster persistence was calculated by Kaplan–Meier estimation from time of first detection of a transmission cluster in a community to the first visit where the transmission cluster was not detected without subsequence re-appearance. The difference between SDSE and *S. pyogenes* was calculated by Cox Proportional Hazards as implemented in survival v3.4.0 and survminer v0.4.9.

### Household transmission network

Transmission networks between households in each community were inferred using a modified version of the model described by ref. 17. and the R packages igraph v1.3.5 for network analysis and ggraph v2.1.0, and scatterpie v0.1.8 for visualisation. Networks were initially inferred for SDSE and *S. pyogenes* separately. Transmission

clusters predicted using WGS were mapped against epidemiological metadata to generate adjacency matrices from which transmission networks were inferred. Each household was represented by a node within the transmission network and unweighted edges (transmission events or links) were drawn between households, or within the same household when unique individuals carried isolates from the same transmission cluster across successive community visits (transmission window between 12 and 44 days). Isolates could be linked to multiple other isolates within the same transmission cluster and respective transmission window. The isolate detected at the earlier community visit/time point was denoted as putative source for the purposes of predicting the contribution of throat and skin carriage/infection to transmission. Edges were drawn for each transmission window and assigned to the latter, 'recipient', community visit. As households were sampled in order over a short window (range 1–4 days) within each community visit, transmission edges were not inferred within the same community visit given the uncertainty in predicting a source. Isolates from individuals without an identified household were excluded.

To determine transmission overlap between SDSE and *S. pyogenes*, the intersection between the transmission networks for SDSE ($G_{sdse}$) and *S. pyogenes* ($G_{pyo}$) at each transmission window ($w$) was taken. Here, households were nodes and with undirected edges ($E$) drawn if they were linked by a transmission event. The percentage of shared edges ($f$) was calculated as:

$$f = \frac{\sum_{w=1}^{n} \left| \left( E_{sdse,w} \cap E_{pyo,w} \right) \right|}{\sum_{w=1}^{n} \left| \left( E_{sdse,w} \cup E_{pyo,w} \right) \right|} \times 100 \tag{1}$$

### Models of independent transmission

A null model of independent transmission was generated for the SDSE and *S. pyogenes* transmission networks by node-label permutation. The household labels of the transmission adjacency matrix generated from the observed SDSE data were permuted over 10,000 iterations and compared to the *S. pyogenes* transmission network inferred from observed data as described above. The number of overlapping edges at each transmission window was calculated, summed for each iteration, and compared to the observed number of shared transmission edges. A one-sided *p*-value testing the hypothesis of SDSE and *S. pyogenes* transmission interference was calculated by the proportion of permutations with shared edges ≤ observed shared edges.

A model of independent inter- and intra-species transmission for household co-occurrence was generated by permutation of positive SDSE and *S. pyogenes* swabs across individuals and households within each community visit while accounting for grouping of isolates from the same transmission cluster within households. Sequenced isolates from the same transmission cluster and household visit (159/609 SDSE and *S. pyogenes* combined, 26%) were collapsed into a single positive result (159 collapsed to 69 positive swabs). Positive SDSE and *S. pyogenes* swabs at each community visit were then permuted across all individuals sampled at that respective community visit. A co-occurrence within a household was counted when SDSE and *S. pyogenes* were present simultaneously in individuals in a household regardless if they were from the same individual or across multiple individuals. After 10,000 iterations, a one-sided *p*-value testing the hypothesis of SDSE and *S. pyogenes* transmission interference at a household level was calculated by the proportion of permutations with co-occurrences ≤ observed co-occurrences.

### Reporting summary

Further information on research design is available in the Nature Portfolio Reporting Summary linked to this article.

## Data availability

The sequence data generated in this study have been deposited in the European Nucleotide Archive under BioProject identifier PRJEB35476. The full de-identified clinical data are available under restricted access for ethical and privacy purposes; reasonable requests for access can be discussed by contacting the corresponding author by email. The processed epidemiological data used to generate these analyses are available in Supplementary Data 1a. *S. pyogenes* sequences have previously been published[17] and are available under BioProjects PRJNA879913 and PRJEB2232. The authors confirm all supporting data have been provided within the article or in supplementary data files. Source data are provided with this paper.

## Code availability

Scripts used to generate the transmission networks and null models are available at https://github.com/OuliXie/SDSE_transmission (https://doi.org/10.5281/zenodo.10852290). Scripts for MGE extraction and classification from the pangenome are updated from a version used to analyse global SDSE and *S. pyogenes* datasets[13] and are available at https://github.com/OuliXie/Strep_MGE_pipeline (https://doi.org/10.5281/zenodo.10852306).

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

## Acknowledgements

We thank the participants, communities, councils, Aboriginal research officers and health centres for their involvement in the original surveillance study. We thank Ross M. Andrews for his role in the original surveillance study. We acknowledge the assistance of the sequencing and pathogen informatics core teams at the Wellcome Sanger Institute, UK where this work was supported by the Wellcome Trust core grants 206194 and 108413/A/15/D. O.X. was supported by the Australian Health and Medical Research Council (NHMRC) postgraduate scholarship (GNT2013831) and Avant Foundation Doctors in Training Research Scholarship (2021/000017). M.R.D. was supported by a University of Melbourne CR Roper Fellowship.

## Author contributions

O.X. worked on study design, analysis, data interpretation and manuscript preparation. M.R.D. and S.Y.C.T. contributed to conception of the project and data interpretation. C.Z., G.T.H., D.J.P., and J.A.L. contributed to transmission model design and data interpretation. J.M.M., M.I.M., A.C.B., P.M.G., B.J.C., J.R.C., and D.C.H. contributed to data collection and curation. S.D.B. contributed to genomic sequencing. All authors contributed to manuscript preparation and review.

## Competing interests

The authors declare no competing interests.

## Additional information

---

[1]Department of Infectious Diseases, University of Melbourne at the Peter Doherty Institute for Infection and Immunity, Melbourne, VIC, Australia. [2]Monash Infectious Diseases, Monash Health, Melbourne, VIC, Australia. [3]School of Computing and Information Systems, University of Melbourne, Melbourne, VIC, Australia. [4]Department of Biostatistics, University of Oslo, Oslo, Norway. [5]Centre for Epidemiology and Biostatistics, Melbourne School of Population and Global Health, University of Melbourne, Melbourne, VIC, Australia. [6]Department of Microbiology and Immunology, University of Melbourne at the Peter Doherty Institute for Infection and Immunity, Melbourne, VIC, Australia. [7]Division of Tropical Health and Medicine, James Cook University, Townsville, QLD, Australia. [8]Wesfarmers Centre for Vaccines and Infectious Diseases, Telethon Kids Institute, University of Western Australia and Perth Children's Hospital, Perth, WA, Australia. [9]Global and Tropical Health Division, Menzies School of Health Research, Charles Darwin University, Darwin, NT, Australia. [10]Faculty of Health, Charles Darwin University, Darwin, NT, Australia. [11]Infectious Diseases Department, Royal Darwin Hospital, Darwin, NT, Australia. [12]Wellcome Sanger Institute, Wellcome Genome Campus, Hinxton, UK. [13]Victorian Infectious Diseases Service, The Royal Melbourne Hospital at the Peter Doherty Institute for Infection and Immunity, Melbourne, VIC, Australia. [14]These authors contributed equally: Mark R. Davies, Steven Y. C. Tong. ✉e-mail: steven.tong@unimelb.edu.au

