## [Peer Review File · Nature Communications]

REVIEWERS' COMMENTS

Reviewer #1 (Remarks to the Author):

The authors have provided cogent and full responses to the earlier critiques. The paper is technically sound even though the design of the original study was not ideal for the questions now being addressed by the authors.

The issue about single colonies being saved from individuals has been mentioned.

-If detection of GAS/SDSE coinfection in the same swabs is not feasible (due to the age of the samples precluding re-plating), did the authors at any stage plan new sampling and a metagenomic approach? It might not require many samples if coinfection is frequent. On the other hand, (noting that the original study did save additional colonies if morphologically distinct) if only 14/558 throat swabs had both GAS and SDSE identified from the original study, could that mean coexistence of SDSE and GAS is in fact rather infrequent?

-In some respects those 14 throat swabs (paired GAS/SDSE isolates) are more interesting than all the others. Did the authors identify any specific shared genomic content between the GAS and SDSE from these 14 swabs compared with all the others?

The authors point out that both homologous recombination and gene transfer account for shared genetic material between GAS and SDSE and that further investigation requires a new study. It would however be interesting for readers to know what the constraints might be for this to happen; whether in specific genotypes, and limited to specific restriction modification systems or other differences in eg CRISPR etc.

Reviewer #2 (Remarks to the Author):

Streptococcus dysgalactiae subspecies *equisimilis* (SDSE) has not been a frequently studied pathogen, and the few studies have shown its increasing importance as a potential pathogen augmenting AHD/RHD especially among poor rural aboriginal populations. Through a longitudinal follow up design and genomic sequencing, the authors studied a total of 294 SDSE and 315 *S. pyogenes* isolates obtained from 4,547 person-consultations within a remote Aboriginal settlement.

SDSE and *S. pyogenes* transmission intersected extensively among households and the observed co-occurrence and transmission links were consistent with independent transmission without inter-species interference. At least one of three near-identical cross-species mobile genetic elements (MGEs) carrying

antimicrobial resistance or

streptodornase virulence genes. The major finding here, that *Streptococcus pyogenes* and (SDSE) commonly interact in their common niches and exchanges MGE including AMR and some virulence genes is a major concern regarding potential to influence the evolution of one or both of these pathogens. In the event of development and deployment of control strategies, including vaccines for either of them, then the epidemiology of the complement would be affected. Thus, it is important that the ecology and genomics of transmission and evolution of both pathogens are well understood.

The authors have elaborately shown that for these 2 pathogens that disproportionately affect the Aboriginal community it would be prudent to integrate SDSE and *S. pyogenes* surveillance and control efforts thus maximizing outcome for resources used.

The methods used in the study including household sampling and longitudinal follow up sampling frame in the population is statistically viable to derive the conclusions made regarding the interactions observed for the 2 pathogens at household level. The use of Whole Genome Sequencing and phylogenetic analysis of important genetic traits within and between the 2 pathogens provides an important basis for concluding that there were independent transmission pathways at household and community level. However, the inference of horizontal gene transfers for AMR and certain virulence genes were equally crucial findings that would account for similarities and variations observed among the 2 pathogens.

What is new?

Authors have shared new additional data on SDSE transmission dynamics and cross-transfers of virulence genes and AMR genes, potentially affecting consideration for vaccine development for both pathogens. The high level of genomic relatedness and sharing of similar niches at individual and household level is an important factor for consideration for any efforts towards management and control of both pathogens which should be jointly prioritized.

What the authors did not tell us?

Whether the horizontal gene transfers between and across the 2 species are dependent on rates of pathogen transmission at household level which would be affected by seasonality, immune status and age of individuals?

Reviewer #2 (Remarks on code availability):

I was able to install and run the code

There is evidence the data sharing in this code is optimally presented and is usable as a resource for the scientific community

Response to reviewer comments

Overlapping *Streptococcus pyogenes* and *Streptococcus dysgalactiae* subspecies *equisimilis* household transmission and mobile genetic element exchange

Reviewer 1

- 1. The issue about single colonies being saved from individuals has been mentioned. If detection of GAS/SDSE coinfection in the same swabs is not feasible (due to the age of the samples precluding re-plating), did the authors at any stage plan new sampling and a metagenomic approach? It might not require many samples if coinfection is frequent. On the other hand, (noting that the original study did save additional colonies if morphologically distinct) if only 14/558 throat swabs had both GAS and SDSE identified from the original study, could that mean coexistence of SDSE and GAS is in fact rather infrequent?**

Unfortunately returning to the communities recruited as part of the original study is no longer practical and would require design of a new study. We are certainly exploring unbiased sequencing approaches for future surveillance studies both in hyperendemic and in lower-incidence urbanised settings. In the context of this current study, development of a metagenomic workflow to allow high sensitivity detection and adequate coverage for transmission inference directly from swabs is beyond the scope of this study. While there were only 15 occasions in 14 individuals where both SDSE and *S. pyogenes* were isolated from the same swab, it cannot be concluded that co-colonisation of the two pathogens is infrequent based on this data. Multiple colonies were only selected if there

were multiple colony variants, and it is expected that SDSE and *S. pyogenes* are indistinguishable morphologically on blood agar.

- 2. In some respects those 14 throat swabs (paired GAS/SDSE isolates) are more interesting than all the others. Did the authors identify any specific shared genomic content between the GAS and SDSE from these 14 swabs compared with all the others?**

While it is interesting that these 15 swabs had colony variants, analysing these as a group compared to all others may be misleading. We demonstrate that very closely related strains form transmission chains within these communities (differ by <8 SNPs for SDSE and <6 SNPs for *S. pyogenes* and >99% shared gene content in both cases). Therefore, these cases represent intersections of *S. pyogenes* (12 transmission clusters totalling 147 isolates) and SDSE (11 transmission clustering totalling 69 isolates) transmission chains rather than unique isolates which co-occur. We found that ϕ 1207.3 was present in two transmission clusters which had members found in co-occurring swabs, ϕ MGAS5005.3 was found in *emm1 S. pyogenes* isolates which co-occurred with SDSE, and the ICE-like element was found in two SDSE transmission clusters which had members in co-occurring swabs.

- 3. The authors point out that both homologous recombination and gene transfer account for shared genetic material between GAS and SDSE and that further investigation requires a new study. It would however be interesting for readers to know what the constraints might be for this to happen; whether in specific genotypes, and limited to specific restriction modification systems or other differences in eg CRISPR etc.**

Thank you for highlighting this interesting area of cross-species interactions. It has previously been hypothesised that CRISPR differences (including acquisition of spacers) may underlie some differences between the two species in MGE profile. However, this

remains unproven. The restriction modification systems of SDSE remain poorly characterised and the repertoire of phage receptors across the two pathogens is unknown. We have added a sentence (lines 345-347) in the discussion to mention some of these unresolved hypotheses.

Reviewer 2

1. What is new?

Authors have shared new additional data on SDSE transmission dynamics and cross-transfers of virulence genes and AMR genes, potentially affecting consideration for vaccine development for both pathogens.

The high level of genomic relatedness and sharing of similar niches at individual and household level is an important factor for consideration for any efforts towards management and control of both pathogens which should be jointly prioritized.

What the authors did not tell us?

Whether the horizontal gene transfers between and across the 2 species are dependent on rates of pathogen transmission at household level which would be affected by seasonality, immune status and age of individuals?

Thank you for your comments and summary of our study. Integrating factors such as immune status with transmission surveillance would be of great interest. The addition of host factors would provide invaluable data regarding host-pathogen interplay and potentially shed light on novel disease control strategies. We did not notice any seasonality in the incidence of SDSE in the original surveillance study. We are unable to disentangle household transmission rates and rates of HGT transfer in this study. The strains making up transmission clusters with cross-species MGEs are found extensively in the communities

in this study. For example, 16/18 households in community 1 had at least one person infected with an SDSE strain carrying a cross-species MGE. Comparing transmission surveillance in a lower disease burden setting while developing methods to allow comparison of rates of HGT across different studies, would be required to analyse the impact of rate of transmission on HGT.